# Gut Microbiota and Oral Contraceptive Use in Women with Polycystic Ovary Syndrome: A Systematic Review

**DOI:** 10.3390/nu16193382

**Published:** 2024-10-04

**Authors:** Jakub Wydra, Katarzyna Szlendak-Sauer, Magdalena Zgliczyńska, Natalia Żeber-Lubecka, Michał Ciebiera

**Affiliations:** 1Department of Endocrinology, Centre of Postgraduate Medical Education, 01-809 Warsaw, Poland; 2Second Department of Obstetrics and Gynecology, Centre of Postgraduate Medical Education, 00-189 Warsaw, Poland; 3Warsaw Institute of Women’s Health, 00-189 Warsaw, Poland; 4Department of Obstetrics, Perinatology and Neonatology, Centre of Postgraduate Medical Education, 01-809 Warsaw, Poland; 5Department of Gastroenterology, Hepatology and Clinical Oncology, Centre of Postgraduate Medical Education, 02-781 Warsaw, Poland

**Keywords:** polycystic ovary syndrome, hyperandrogenism, gut microbiota, combined oral contraceptives

## Abstract

**Background:** Polycystic ovary syndrome (PCOS) is one of the most prevalent endocrine syndromes affecting women at reproductive age. With increasing knowledge of the role of the microbiota in the pathogenesis of PCOS, new management strategies began to emerge. However, data on the impact of established treatment regimens, such as metformin and oral contraceptive agents, on the gut microbiota composition are scarce. This study aimed to evaluate the specificity of the gut microbiota in women with PCOS before and after treatment with oral contraceptives. **Methods:** We have systematically searched the following databases: PubMed/MEDLINE, Scopus, Web of Science and Google Scholar. The last search was performed on 13 May 2024. We included only full-text original research articles written in English. The risk of bias was assessed using a modified version of the Newcastle–Ottawa Scale. **Results:** The above described search strategy retrieved 46 articles. Additionally, 136 articles were identified and screened through Google Scholar. After removing duplicates, we screened the titles and abstracts, resulting in three eligible articles constituting the final pool. They were published between 2020 and 2022 and are based on three ethnically distinct study populations: Turkish, Spanish and American. The studies included a total of 37 women diagnosed with PCOS and using OCs. **Conclusions:** OC treatment does not seem to affect the gut microbiota in a significant way in patients with PCOS in short observation. Well-designed randomized controlled studies with adequate, unified sample size are lacking.

## 1. Introduction

Polycystic ovary syndrome (PCOS) is one of the most prevalent endocrine syndromes affecting women at reproductive age. The diagnosis of PCOS should be made in the presence of two of clinical or biochemical hyperandrogenism, ovulatory dysfunction and polycystic ovaries on ultrasound or elevated anti-mullerian hormone (AMH) levels, after excluding other potential causes. In adolescents, both hyperandrogenism and ovulatory dysfunction are required for the diagnosis [1,2]. Hyperandrogenism is mostly of ovarian origin, driven by gonadotropic-stimulating hormone and luteinizing hormone-stimulating ovarian theca cells to secrete androgens, which in turn halt follicular growth and cause ovulatory dysfunction [3]. Insulin resistance, another inherent feature of PCOS, possibly regulated by endogenous opiates, further exacerbates the effect of hyperandrogenemia [3,4]. Chronic hormonal imbalance, low-grade inflammation and insulin resistance associated with PCOS are accompanied by an increased risk of comorbidities, such as endometrial cancer, and metabolic syndrome, including type 2 diabetes. It significantly affects the general well-being of patients and reduces their quality of life [5]. Thus, numerous potential molecular pathways are currently being researched in the hope of elucidating new treatment targets in patients with PCOS.

Numerous environmental factors, such as nutrition, socioeconomic status and environmental pollution, may contribute to the pathophysiology of PCOS [6]. Recently, numerous authors advocated a connection between PCOS and the gut microbiota, suggesting its role in the development of the syndrome and a possible treatment target [7,8]. In the gut microbiome, the majority of bacterial species belong to the phyla *Firmicutes* and *Bacteroidetes*, with fewer belonging to the phyla *Actinobacteria*, *Proteobacteria*, *Fusobacteria* and *Verrucomicrobia*. Dysbiosis, defined as the disturbance of the gut microbiota, is increasingly recognized as an indicator of disease occurrence [9]. Numerous studies revealed a significant difference in the gut microbiota of women with PCOS when compared to healthy controls [7,8,10,11,12,13,14,15,16,17,18,19,20,21,22,23,24]. According to those studies, PCOS was associated with a decrease in microbial diversity, the population of beneficial bacteria, including *Lactobacillus* and *Bifidobacterium*, and an increase in pathogenic genera, such as *Escherichia* and *Shigella* [7,8]. One hypothesis suggested that poor diet induced gut dysbiosis, which in turn led to increased intestinal permeability and the passage of lipopolysaccharides of Gram-negative bacteria into the bloodstream, inducing immune system response [25,26]. Chronic immune system activation interferes with insulin receptors, causing hyperinsulinemia and hyperandrogenism [27,28,29,30], which is the main endocrine anomaly in PCOS and one of the diagnostic criteria [2].

The treatment of hyperandrogenism in PCOS is crucial for symptomatic patients with androgenic skin symptoms, like hirsutism estimated according to the modified Ferriman–Gallwey (mFG) score, acne and alopecia. Oral contraceptive (OC) agents were established as the first-line treatment and most commonly used option in most women with PCOS who did not want to conceive [2,31,32]. OCs were shown to affect androgen synthesis and metabolism, including both ovarian production, together with stimulating sex hormone-binding globulin synthesis, thus ameliorating skin androgenic symptoms and regulating menstrual bleeding [31,32].

With increasing knowledge of the role of the microbiota in the pathogenesis of PCOS, new management strategies began to emerge. Potential therapeutic options, such as prebiotics, probiotics, synbiotics and fecal microbiota transplants, were described to normalize altered microbial composition and potentially alleviate PCOS symptoms [33,34].

However, data on the impact of treatment regimens recommended by the most recent PCOS guidelines [2], such as metformin and oral contraceptive agents, on the gut microbiota composition are scarce and the available studies were performed on small patient groups. Basing on the systematic review from 2023 that included mostly studies mainly on patients with type 2 diabetes, metformin’s effect on gut microbiota diversity and specific genera seems inconsistent across populations [35]. Data on the effect of OC on the gut microbiome in healthy women are also limited and indicate rather minor changes in gut microbiota diversity and abundance of bacterial taxa [36].

Thus, this study aimed to evaluate the specificity of the gut microbiota in specified group of patients: women with PCOS before and after treatment with OCs.

## 2. Material and Methods

The literature search for articles regarding polycystic ovary syndrome, microbiota and contraception was performed using the following databases: PubMed/MEDLINE, Scopus, Web of Science and Google Scholar. The search strategy was properly adapted to each database. Moreover, the authors hand-searched the references of eligible studies in order to obtain a full view on the topic. Details on the search strategy are summarized in Table 1.

The last search was performed on 13 May 2024 and there were no restrictions on the date of publication. This study aimed to obtain the following PICOS data:-Population: women with polycystic ovary syndrome;-Intervention: treatment with oral contraceptives (OCs);-Comparison: women with PCOS before treatment with OCs;-Outcomes: comparison of gut microbiota composition in women with PCOS before and after treatment with OCs.

Only full-text original research articles written in English were considered eligible for analysis. We excluded studies from our analysis according to the following criteria: reviews, editorials, opinions or letters, case reports or case series, conference papers, abstracts, unavailability of the primary outcome of interest.

This review followed the Preferred Reporting Items for Systematic Reviews and Meta-Analyses (PRISMA) guidelines [37]. No institutional review board approval was required for this study.

## 3. Data Extraction

Using a custom-built data extraction form, we sought the following information: the authors, year and country of origin, type of study, the main objective, population characteristics, diagnostic methods used and the results. Two study authors extracted the above mentioned data from the selected full-text articles, while the third author double-checked their accuracy. No quantitative analyses were performed due to small sample sizes and a low number of studies with a relatively short follow-up.

## 4. Risk of Bias

Two authors individually assessed the risk of bias in the selected studies using a modified version of the Newcastle–Ottawa Scale adapted by the authors for the needs of this systematic review (Appendix A). Any disagreements were resolved through discussion and consensus with all study authors.

## 5. Results

The above described search strategy retrieved 46 articles. After the automated deletion of duplicates with EndNote 20 (Clarivate Analytics, London, UK), we manually removed the remaining duplicates and carefully reviewed the abstracts and full texts. Additionally, 136 articles were identified and screened through Google Scholar. Moreover, in order to avoid omitting any article relevant to the topic, we reviewed the papers that have cited the identified articles. Titles, abstracts and full-text versions of the research papers were assessed independently by two authors. The study selection process is summarized in the PRISMA flow chart (Figure 1).

As a result of the search process, three eligible articles were identified and constituted the final pool. They were published between 2020 and 2022 and were based on three ethnically distinct study populations: Turkish, Spanish and American. The studies included a total of 37 women diagnosed with PCOS and using OCs [38,39,40]. The largest group included 17 patients [40]. PCOS was diagnosed based on various criteria. Two studies focused only on overweight patients or those with any degree of obesity [39,40], and one on a group of women with normal body weight or who were overweight [38]. Two of the studies focused on the adolescent population [38,39]. All studies included only women without a diagnosis of diabetes. The patients used OCs with various compositions. All research groups assessed the microbiome using 16S rRNA gene amplicon sequencing. A summary of the characteristics of research articles included in the review is provided in Table 2.

## 6. Synthesis of Results and Discussion

Alterations in the gut microbiota in patients with PCOS were consistently described across numerous studies despite various study designs and conditions [7,8,10,11,12,13,14,15,16,17,18,19,20,21,22,23,24,41]. The specific changes, however, remain equivocal. Both marked the decrease or lack of change in diversity indices with alterations in evenness and phylogenetic abundance being described [23,42,43]. Nevertheless, dysbiosis is believed to be one of the components in the pathophysiology of the onset and progression of PCOS, because of its link with hyperandrogenism [23,42,43].

Clinical hyperandrogenism and testosterone concentrations were shown to be negatively correlated with alpha diversity [43]. It has been shown that sex regulates the maturation of the gut microbiome after puberty [42,44]. Furthermore, it has been shown that shifts in androgens and estrogens can change the composition of the gut microbiota in animal models [45,46]. This suggests that differences in the gut microbiome in males and females are at least partially caused by steroid hormone levels. Moreover, altered gut microbiota in obese adolescents with PCOS versus obese adolescents without PCOS was described and the changes related to metabolic markers and testosterone [23]. These studies suggest that hyperandrogenism may have a major impact on the gut microbiota in women with PCOS.

Both prospective studies assessed the baseline status of the gut microbiota in women with PCOS compared to matched healthy controls [38,40]. Eyupoglu et al. showed no difference in alpha and beta diversity, and the number of species between PCOS and controls, whereas Garcia-Beltran et al. confirmed lower alpha diversity in a PCOS group [38,40]. Eyupoglu et al., who analyzed operational taxonomic units (OTUs), showed lower counts in obese PCOS than in obese controls [40]. As for species, Eyupoglu et al. noted abundant *Ruminococcaceae* in PCOS, and Garcia-Beltran et al. reported abundant Family XI and depleted *Prevotella* and *Senegalimassilia* in PCOS [38,40]. The differences in the gut microbiota of patients with PCOS vary across numerous observational studies. However, a meta-analysis of 19 human studies showed that patients with PCOS had a lower Chao index, Shannon index and OTU counts with higher relative abundance of *Bacteroidaceae* compared to healthy controls. However, no significant differences occurred as regards other phyla and at the family or genus level [41]. Three previous studies analyzed overweight and obese patients with PCOS and reported decreased alpha and beta diversity compared to the healthy control group. The authors suggested significant differences between obese and non-obese subjects with PCOS [23,24,47]. The inconsistency of diversity indices might be explained by the small sample sizes of the abovementioned studies, as well as regional and dietary differences, as populations differed in terms of the country or even continent of origin, ethnicity, age and BMI.

The baseline microbiome was also correlated with the phenotype of PCOS patients. Eyupoglu et al. found that OTU counts and the number of species were negatively correlated with carbohydrate metabolism parameters, whereas *Ruminococcaceae* were positively correlated with modified Ferriman–Gallwey (mFG) score at baseline [40]. This is consistent with the previous study in which letrozole-induced PCOS resulted in the greater relative abundance of Ruminococcaceae genus in pubertal female mice [43]. Conversely, in a Mendelian randomization study, Sun et al. suggested that *Ruminococcus*, a member of the family *Ruminococcaceae*, had a beneficial, protective effect against PCOS and was associated with improved insulin sensitivity in obese patients [48]. According to Garcia-Beltran et al., Family XI was negatively correlated with high-molecular-weight adiponectin (HMW-adiponectin) and positively correlated with hepato-visceral fat. *Prevotellaceae* were positively correlated with SHBG, LDL, HMW-adiponectin and negatively with us-CRP, whereas *Senegalimassilia* were positively correlated with testosterone, SHBG, FAI us-CRP and hepato-visceral fat. Similar differences in the gut microbial composition in overweight and obese women with PCOS compared to healthy controls were previously observed, and total and free testosterone were correlated with diversity measures [23,24,47]. Zhou et al. observed that insulin-resistant PCOS was associated with decreased alpha diversity, increased proinflammatory *Bacteroides* and decreased *Prevotellaceae* [47]. Jobira et al. found that HOMA-IR was significantly associated with family *Lachnospiraceae* and *Veillonellaceae* [23]. Contrary to this, Sun et al. suggested that the genus *Sellimonas* belonging to the family *Lachnospiraceae* might have protective effects against PCOS [48].

Tayachew et al. [39] performed a methodologically different study—a secondary analysis of cross-sectional studies, the results of which also require careful interpretation. Gut microbiome profiles were assessed, among others, in adolescent, obese patients with PCOS receiving OCs. Free testosterone levels were negatively correlated with alpha diversity and only HOMA-IR was negatively correlated with the abundance of *Ruminococcaceae* family in patients with PCOS, despite a previously described association with testosterone levels in PCOS [19].

Regarding the impact of treatment on the microbiome, Eyupoglu et al. noted a trend for a decrease in the relative abundance of the *Actinobacteria* phylum after OC treatment, which was significant only in the obese PCOS subgroup. During the follow-up, a decrease in BMI (body mass index) and WHR (waist-hip ratio) was noted, which might have potentially influenced the results. WHR was also described to influence gut microbiota composition and be significantly associated with the *Firmicutes/Bacteroidetes* ratio and the family *Bacteroidaceae* [23]. Garcia-Beltran et al. found the reduced abundance of the *Prevotella* genus in OC group. Previous studies describing the relation between *Prevotella* and PCOS were inconsistent [23,47]. In the analysis of cross-sectional studies, Tayachew et al. found higher *Pseudobutyrivibrio* belonging to the *Firmicutes* phylum in women with PCOS taking OCs. The abundance of *Pseudobutyrivibrio* was reported to significantly decrease after 8 weeks of a low-energy diet [49]. This is in line with previous reports describing a positive association between the relative abundance of *Pseudobutyrivibrio* and circulating estrogen [50,51], as such calorie restriction and weight loss were more commonly found to be associated with hypogonadism.

Obesity, particularly visceral obesity, is often associated with PCOS [52]. Various studies have shown that obesity may significantly alter the gut microbiome. According to a recent meta-analysis, at the phylum level, there was a significant increase in Firmicutes. At the genus level, lower relative proportions of *Bifidobacterium* and *Eggerthella* were observed, while higher levels of *Escherichia-Shigella, Eubacterium, Fusobacterium, Prevotella* and *Streptococcus* were found. Additionally, there was a trend, although not statistically significant, towards lower alpha diversity [53]. However, in the analyzed studies, the majority of patients were also overweight or obese. In the study authored by Garcia-Beltran et al., patients with normal body weight were included, but the average BMI in the PCOS group was still 25 kg/m^2^, compared to 22 kg/m^2^ in the healthy control group [38]. The above described microbiome changes, commonly associated with excess body weight, could explain the finding of lower alpha diversity in the PCOS group [53]. Furthermore, current guidelines recommend physical activity for all PCOS patients, which also seems to affect the gut microbiota [2,54,55,56]. Garcia-Beltran et al. and Eyupoglu et al. both recommended regular physical activity [38,40]. Tayachew et al. used an accelerometer to measure physical activity, making the control and study groups comparable [39]. However, the authors did not report any microbiome comparisons related to this aspect.

The study by Eyupoglu et al. had a relatively short 3-month OC treatment, which did not contribute to any significant alterations in the gut microbiota. However, the patient group was simultaneously subjected to lifestyle changes, and followed a standardized 3-day meal plan, which could have also impacted the composition of the gut microbiota. Moreover, the study included only overweight and obese patients with PCOS, used a single form of OC, and encompassed only adult patients. Thus, the results by Eyupoglu et al. cannot be extrapolated to subjects with normal BMI and different components of OC and adolescents. Similarly, Tayachew et al. recruited obese, adolescent PCOS patients with a relatively longer follow-up period. However, no healthy control group was included and different OC formulations were probably used. Furthermore, the microbiota composition of the OC group might have differed to the one in the untreated group prior to OC initiation. Conversely, Garcia-Beltran et al. implemented strict inclusion criteria, an even longer follow-up period of 1 year, excluded obese subjects, and used a different composition of OC, which again makes the direct comparison and synthesis of the results difficult.

Garcia-Beltran et al. [38] described that changes in visceral fat mass and non-biochemical hyperandrogenism were one of the main contributors to gut microbiota alterations, although both hyperandrogenism and hepato-visceral fat may cause gut dysbiosis. This underlines the fact that the normalization of fat mass in PCOS may not only alleviate the clinical manifestations of the disorder but also restore the gut microbiota. Garcia-Beltran et al. did not, however, include obese PCOS patients in the study. Eyupoglu et al. observed a decrease in *Actinobacteria* abundance in the obese subgroup, which did not show a correlation with BMI or androgen parameters despite significant weight loss. Although *Actinobacteria* is a less abundant phylum in the gut microbial composition, its relative abundance was described to be increased in patients with PCOS and in obese individuals by several authors with certain inconsistency [10,23,57,58,59]. However, such changes were not observed in obese PCOS subjects by Tayachew et al. [39].

The shift in circulating androgen levels in PCOS after OC therapy is probably insufficient to impact the gut microbiota as observed in hypogonadal castrated mice [60] and as reported by Harada et al. [61]. It is plausible that androgen shifts in human patients treated for PCOS were less evident in comparison with animal models due to the lack of restrictive diet, which was implemented in mice models [62]. In another study by Eyupoglu et al., OC therapy contributed to a decrease in circulating gut microbiome metabolite, i.e., trimethylamine N-oxide, which was correlated with biochemical hyperandrogenism. However, the composition of the gut microbiota was not investigated [63]. Moreover, higher levels of estradiol were associated with a greater diversity of the gut microbiota [64]. The dose and treatment duration might have been insufficient to observe any significant changes in the gut microbiota caused by the shift in androgens, as it seems to be an indirect effect as contrasted with SPIOMET (spironolactone, pioglitazone and metformin) treatment, which seemed to have directly modulated the gut microbiota during a short treatment period. It is unclear if ethinyloestradiol and estradiol have a similar association with the gut microbiota. Conversely, the withdrawal of letrozole treatment in the pubertal PCOS mice model resulted in a recovery of gut microbiota diversity only after 2 months, suggesting that increased androgen levels during puberty might lead to the development of PCOS, and that the normalization of androgenemia might improve reproductive and metabolic parameters in PCOS [65]. Since sex steroids serve as a carbon source for certain gut microbiota, circulating androgen levels might be regulated by modulating the abundance of androgen-metabolizing bacteria, thus making such probiotics potential alternatives to OC therapy in women with PCOS [66].

The presented studies were low to medium quality in terms of the risk of bias (Appendix A). However, the main limitation of our review was the scarcity of available data. The applied search retrieved only three papers with small sample sizes and relatively short treatment periods; thus, final conclusions cannot be drawn at the moment. The study groups differed significantly in terms of ethnicity, age, BMI and even the disease diagnosis criteria used. Moreover, patients received significantly different interventions—not only different in terms of OC components, but they were also subjected to lifestyle changes in some studies. It remains uncertain whether OC therapy merely ameliorates biochemical and clinical hyperandrogenism without significantly impacting the gut microbiota and the pathogenesis of PCOS, and if probiotics might potentially have a more beneficial effect on the course of PCOS, affecting underlying mechanisms such as low-grade inflammation, together with a significant androgen level reduction.

## 7. Conclusions

To our knowledge, this is the first systematic review of articles assessing gut microbiota in women with PCOS before and after treatment with OC. Basing on the scarce available data, OC treatment seems to affect the gut microbiota in a negligible way in patients with PCOS in a short observation. However, well-designed randomized controlled studies with adequate, unified sample size are lacking.

## Figures and Tables

**Figure 1 nutrients-16-03382-f001:**
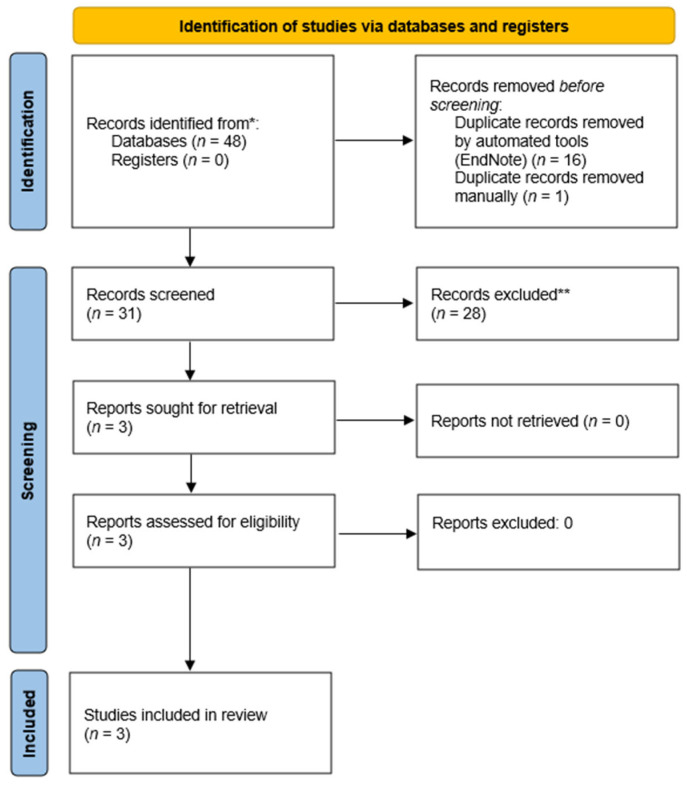
PRISMA flow chart.

**Table 1 nutrients-16-03382-t001:** Search strategy for used databases.

Database	Number of Results	Search Strategy
PubMed	9	(“Microbiota”[Mesh] OR microbiot* OR microbiom* OR microfilm* OR flora OR microflora OR microorganism* OR “high-throughput nucleotide sequencing”[Mesh] OR NGS OR (next AND generation AND sequencing) OR (shotgun AND metagenomic*) OR “RNA, ribosomal, 16S”[Mesh] OR (16S AND rrna)) AND (“polycystic ovary syndrome”[Mesh] OR polycystic OR pco OR pcos) AND (“contraceptives, oral, hormonal”[Mesh] OR “contraceptives, oral, combined”[Mesh] OR (oral AND contracept*) OR (combined AND contracept*))
Scopus	27	TITLE-ABS-KEY(microbiot* OR microbiom* OR microfilm* OR flora OR microflora OR microorganism* OR “high-throughput nucleotide sequencing” OR NGS OR (next AND generation AND sequencing) OR (shotgun AND metagenomic*) OR “RNA, ribosomal, 16S” OR (16S AND rrna)) AND TITLE-ABS-KEY(“polycystic ovary syndrome” OR polycystic OR pco OR pcos) AND TITLE-ABS-KEY((oral AND contracept*) OR (combined AND contracept*))
Web of Science	12	TS = (microbiot* OR microbiom* OR microfilm* OR flora OR microflora OR microorganism* OR “high-throughput nucleotide sequencing” OR NGS OR (next AND generation AND sequencing) OR (shotgun AND metagenomic*) OR “RNA, ribosomal, 16S” OR (16S AND rrna)) AND TS = (“polycystic ovary syndrome” OR polycystic OR pco OR pcos) AND TS = ((oral AND contracept*) OR (combined AND contracept*))
Google Scholar	139	(microbiot* OR microbiom* OR microfilm* OR flora OR microflora OR microorganism* OR “high-throughput nucleotide sequencing” OR NGS OR (next AND generation AND sequencing) OR (shotgun AND metagenomic*) OR “RNA, ribosomal, 16S” OR (16S AND rrna)) AND (“polycystic ovary syndrome” OR polycystic OR pco OR pcos) AND ((oral AND contracept*) OR (combined AND contracept*))

**Table 2 nutrients-16-03382-t002:** Results.

Authors, Year, Country of Origin	Study Design	Aim	Analyzed PCOS Population	Duration of Treatment	Treatment	Control Group	Effect of OC Treatment in PCOS Group	Diversity at Baseline	Species at Baseline	Effect of Treatment on Microbiota in PCOS
Eyupoglu et al. 2020 Turkey [40]	Prospective observational study	To assess if gut mibrobiota is altered in PCOS and to determine the impact of OCs	*n* = 17 on OCs • median age 20 years • median BMI 29.6 kg/m^2^ (all between 25–40 kg/m^2^) • glucose-tolerant • diagnosis based on the Rotterdam criteria	3 months between April and December 2018	Dienogest and ethinyloestradiol (2 mg + 0.03 mg) + healthy diet + a minimum of 150min/week of moderate intensity physical activity	*n* = 15 age- and BMI-matched healthy controls	• lower BMI, WHR, TT, FAI • higher fasting insulin, HOMA-IR	• no difference in alpha and beta diversity, number of species between PCOS and controls	• *Ruminococcaceae* enriched in PCOS • OTU lower in obese PCOS than obese controls • OTU counts higher in overweight than obese irrespective of PCOS status	• no difference in OTU counts and alpha and beta diversity after treatment • abundance of *Ruminococcaceae* in PCOS did not change after OC • trend for a decrease in the relative abundance of Actinobacteria phylum after OC, significant in the obese PCOS subgroup
Garcia-Beltran et al. 2021 Spain [38]	Randomized controlled trial	To assess the gut microbiota composition of girls with PCOS without obesity and compare the effects of OCs or SPIOMET	*n* = 12 on OCs, *n* = 11 on SPIOMET • median age 15 years • mean BMI 25 kg/m^2^ (all without obesity) • glucose-tolerant • diagnosis based on hirsutism >8 mFG score, oligomenorrhea >45 days, gynecological age > 2.0 years	1 year between December 2015 and October 2019	• levonorgestrel and ethinyloestradiol (100 mg + 0.02 mg) OR • SPIOMET = spironolactone 50 mg/d, pioglitazone 7.5 mg/d, and metformin 850 mg/d	*n* = 31 age-matched healthy controls	• higher BMI, Z-score, SHBG, us-CRP, fat mass, abdominal fat measured by DXA, subcutaneous fat measured by MRI • lower TT, FAI	• lower alpha diversity in PCOS • differences in community structure regarding beta-diversity • baseline alpha diversity measures correlated with SHBG and FAI and ALT	• in PCOS abundant Family XI and depleted *Prevotellaceae* • *Prevotella* and *Senegalimassilia* depleted in PCOS	• no change in alpha and beta diversity, *Family XI*, *Prevotellaceae*, and *Senegalimassilia* after OC • OC reduced the abundance of genus *Prevotella*
Tayachew et al. 2022 USA [39]	Secondary analysis of 3 separate cross-sectional studies	To assess gut microbiome profiles, serum metabolomics, hormone levels and metabolism in adolescents with PCOS and obesity with and without OC treatment	*n* = 8 on OC • mean age 15.5, • mean BMI 32.5 kg/m^2^ (>90th centile)• without diabetes, • diagnosis of PCOS based on National Institute of Health criteria—adolescent adaptation	>6 months median 8 months (range 6–24)	COC	*n* = 21 age, race, ethnicity, age of menarche and BMI-matched untreated with PCOS	• lower mFG score, free testosterone and FAI • higher SHBG and platelets	unavailable	unavailable	• no difference in alpha and beta diversity • no differences at the phylum or family level between groups • at the genus level the %RA of *Pseudobutyrivibrio* higher in OC group

%RA—relative abundance, ALT—alanine aminotransferase, BMI—body mass index, COC—combined oral contraception, CRP—C-reactive protein, DXA—dual-energy X-ray absorptiometry, FAI—free androgen index, HMW-adiponectin—high-molecular-weight adiponectin, HOMA-IR—homeostatic model assessment for insulin resistance, LDL—low-density lipoprotein, mFG—modified Ferriman–Gallwey score, MRI—magnetic resonance imaging, OC—oral contraceptives, OGTT—oral glucose tolerance test, OTU—operational taxonomic unit, PCOS—polycystic ovary syndrome, SHBG—sex-hormone binding globulin, SPIOMET—spironolactone, pioglitazone and metformin, TT—total testosterone WHR—waist-to-hip ratio.

## Data Availability

The original contributions presented in the study are included in the article. Further inquiries can be directed to the corresponding author.

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
