# Peer review of "Gut Microbiota and Oral Contraceptive Use in Women with Polycystic Ovary Syndrome: A Systematic Review"

_nutrients, 2024, doi:10.3390/nu16193382_

Round 1
Reviewer 1 Report
Comments and Suggestions for Authors
The paper is well designed and presented and the question is interesting. The problem is that in the end they only considered 3 papers out of the initial 136 and perhaps it is too few to carry out a "systematic" review. Perhaps the term "narrative" review would be more appropriate.
Author Response
Dear Reviewer,
Thank you for your insightful assessment and constructive feedback. We appreciate your comments regarding the number of studies included in the final analysis.
We would like to clarify that the term we have used "systematic review" refers to the methodology employed during the search and screening process, as guided by PRISMA guidelines, rather than the number of studies ultimately included. Even when only a few studies meet the inclusion criteria, the process remains systematic. As an example, the prestigious Cochrane Library published a systematic review where no studies were found after the search, yet it was still categorized as a systematic review: https://www.ncbi.nlm.nih.gov/pmc/articles/PMC7133547/.
We hope this addresses your concern, and we are grateful for your thoughtful review of our work.
Reviewer 2 Report
Comments and Suggestions for Authors
Review of "Gut microbiota and Oral Contraceptive use in women with Polycystic Ovary Syndrome: a systematic review." (nutrients-3212524).
This review article focuses on the change in gut microbiota before and after oral contraceptive use in women with PCOS. The authors revealed that the studies themselves were few, and that the results showed no change in gut microbiota before or after OC use.
This study was well written in terms of background and method, and potentially interesting. This reviewer has several questions.
1. Table 1. What do numbers such as PubMed – “9” or Scopus – “27” mean?
2. According to the Introduction, metformin is also a recommended treatment regimen in the latest PCOS guidelines. How about the impact of metformin on the gut microbiota in women with PCOS or animal model?
3. Are there any previous studies on the impact of OC usage on gut bacteria?
Reviewer 3 Report
Comments and Suggestions for Authors
Comment on “Gut microbiota and Oral Contraceptive use in women with Polycystic Ovary Syndrome: a systematic review”
This is an interesting review exploring the link between oral contraceptive use and Gut microbiota in women with PCOS. The topic is update, but the review needs some correction before publication.
In the introductive paragraph please provide an update and complete definition of PCOS and provide an overview of its pathogenetic mechanisms, citing and discussing the following articles:
- Joham AE, Norman RJ, Stener-Victorin E, Legro RS, Franks S, Moran LJ, Boyle J, Teede HJ. Polycystic ovary syndrome. Lancet Diabetes Endocrinol. 2022 Sep;10(9):668-680. doi: 10.1016/S2213-8587(22)00163-2. Epub 2022 Aug 4. Erratum in: Lancet Diabetes Endocrinol. 2022 Nov;10(11):e11. doi: 10.1016/S2213-8587(22)00281-9. PMID: 35934017.
- Lanzone A, Fulghesu AM, Fortini A, Cutillo G, Cucinelli F, Di Simone N, Caruso A, Mancuso S. Effect of opiate receptor blockade on the insulin response to oral glucose load in polycystic ovarian disease. Hum Reprod. 1991 Sep;6(8):1043-9. doi: 10.1093/oxfordjournals.humrep.a137482. PMID: 1666896.
I suggest adding two paragraphs before the key discussion on the relationship of gut microbiota, COC and PCOS:
- a paragraph describing the known relationship between gut microbiota and PCOS, underlying the influence of obesity and hyperandrogenism.
- a paragraph summarizing the known influence of COC on gut microbiota.
I also suggest to modify the table 2 making it more concise, as in its present form it is dispersive and difficult to read.
Finally add a conclusive paragraph summarizing the novelty of your review
Reviewer 4 Report
Comments and Suggestions for Authors
The authors have reported the literature studies related to the association of treatment of PCOS with OC and changes of gut microbiota and conclude that treatment with OC does not affect the gut microbiota after short term evaluation
Comments:
- the literature search is opinable and the search strategy is mainly not related to the topic. I suggest considering a review of literature including only the terms PCOS and contraceptives and microbiota or microbiome and I’m sure that the result would be the same.
-lines 29-30 The authors report an effect of OC on adrenal androgen production. Usually OC affect protein binding of adrenal steroids but they do not a have a direct action on adrenal androgens that are regulated by ACTH
-The concept that contraceptives improve menstrual dysfunction is also opinable: women with PCOS have oligo or amenorrhea and this finding is usually worsened after withdrawal of OC.
-Ref 34 rightly reports the importance of lifestyle interventions and of the treatment with spironolactone and/or metformin. The authors should report possible lifestyle interventions and associated treatments used in the 3 studies involving OC. They should also better discuss the implication of microbiota both in obese women with- or without PCOS
-In conclusion the study is of interest but should be rewritten deleting the actual search strategy for databases and considering only the few studies evaluating gut microbiota in women with PCOS treated with OC, considering that contraceptives are not appropriate in women with PCOS and obesity and insulin resistance or hypertension. These associations maybe are the reason for a lack of improvement of microbiota is related to side effects of OC
Round 2
Reviewer 2 Report
Comments and Suggestions for Authors
The authors revised well. This reviewer has no further comments.
Reviewer 4 Report
Comments and Suggestions for Authors
no further comments